# Impact of COVID-19 on Czech Dentistry: A Nationwide Cross-Sectional Preliminary Study among Dentists in the Czech Republic

**DOI:** 10.3390/ijerph18179121

**Published:** 2021-08-29

**Authors:** Jan Schmidt, Eliska Waldova, Stepanka Balkova, Jakub Suchanek, Roman Smucler

**Affiliations:** 1Department of Dentistry, Charles University, Faculty of Medicine in Hradec Kralove and University Hospital Hradec Kralove, 500 05 Hradec Kralove, Czech Republic; Jan.Schmidt@lfhk.cuni.cz; 2Wald Pharmaceuticals, s.r.o., Detska 37, 100 00 Prague, Czech Republic; eliska.waldova@enzymel.cz; 3Czech Dental Chamber, Slavojova 270/22, 128 00 Prague, Czech Republic; stepabal@gmail.com (S.B.); smucler@dent.cz (R.S.)

**Keywords:** COVID-19, dentistry, pandemic, dentist, protective equipment

## Abstract

This work evaluates the impact of the COVID-19 pandemic on Czech dentistry from March 2020 to March 2021. The assessment was based on questionnaires filled out by 3674 Czech dentists representing 42.6% of practicing dentists in the country. During March–May, 2020 (the first COVID-19 wave), 90.7% of dental practices remained open; however, only 22.8% of the practices continued to operate with no changes, 46.5% had fewer patients, 21.4% treated only acute cases, and 3.8% were closed. During September 2020–May 2021 (the second wave of COVID-19), 96.1% of dental practices remained open, 60.8% operated with no changes, 34.5% had fewer patients, 0.8% treated only acute cases, and 0.5% were closed. The reasons leading to the closure of Czech dental practices during the whole pandemic were a shortage of personal protective equipment (50.5%), a COVID-19 outbreak in the workplace (24.5%), fear of a possible self-infection (24.0%), and quarantine (20.5%). The time range of Czech dental practices closure during the whole pandemic was: 1–2 weeks (49.9%), 2–4 weeks (21.2%), and >1 month (0.8%). The greatest professional difficulties of Czech dentists during the pandemic were crisis operating management (55%), health safety and hygiene concerns (21%), shortage of personal protective equipment (21%), and difficulty working with the protective equipment (15%). In addition, 47.3% of dentists also observed a declining interest in preventive dental care, and 16.9% of them observed worse oral care of patients. These results show that despite the lack of protective equipment, dental care was maintained throughout the pandemic. Additionally, the pandemic negatively affected the patients’ approach to dental care, indicating a deterioration in oral health as a possible delayed outcome of the COVID-19 pandemic.

## 1. Introduction

Coronavirus disease 2019 (COVID-19) is a contagious disease caused by severe acute respiratory syndrome coronavirus 2 (SARS-CoV-2). Symptoms of COVID-19 are variable, ranging from mild to deadly, including fever, cough, headache, fatigue, loss of smell and taste, breathing difficulties, respiratory failure, or acute inflammatory response (cytokine storm) [1,2,3,4,5,6]. The human-to-human transmission is caused by respiratory droplets, either by being inhaled or deposited on mucosal surfaces and via direct contact of mucous membranes, such as oral, nasal, or eye [7]. It was first reported in Wuhan City, Hubei Province, China, in December 2019 [8]. Since then, COVID-19 has become a worldwide major health concern, and on 11 March 2020, it was declared a pandemic disease [9].

In the Czech Republic, the first laboratory-confirmed case of COVID-19 was announced on 1 March 2020. A state of emergency was declared as of 12 March 2020. The emergency was later extended until 18 May 2020. From 16 March 2020, free movement was limited with the exception of travel to and from work, and essential trips, including grocery shopping, trips to the pharmacy, or aiding elderly members of the family. The key measures issued by the Czech government are described in Table 1 [10,11,12]. Due to these early precautions, the pandemic was mild in the Czech Republic during the spring of 2020, with a cumulative number of 79 COVID-19 positive cases per 100,000 people at the end of the first state of emergency [13]. During late May and early June 2020, the obligation of social distancing and wearing masks was abolished. This period, from March to May 2020, is also called the “first wave” of COVID-19. For the rest of the summer, the number of new cases was minimal. However, in September, new COVID-19 cases began to rise again. A state of emergency was declared again on 5 October 2020 and was to remain in force until 11 April 2021. The pandemic was most severe from September to March and began to subside in April. At the end of the second emergency, the cumulative number of cases per 100,000 people was 14,308 [13]. This period from September to March 2020 is also called the ‘second wave’ of COVID-19.

Healthcare workers were considered particularly vulnerable to the pandemic all around the world, especially those whose work is in close contact with mucus and saliva droplets. Due to their work settings, dentists are one of the most endangered professional groups. At the beginning of the pandemic, the Czech Republic and most European countries lacked protective equipment, even for health professionals. Regardless of the pandemic, according to good medical practice and Czech law, it is the duty of each health care provider to limit the transmission of infectious diseases as much as possible. The same has been true during the pandemic with no exception, leaving health care providers in unprecedented medical and, eventually, legal insecurity. In the case of dentists, the culpable transmission of the disease was at stake. Such conditions were not unique to the Czech Republic and were also faced by health care workers in other European countries [14,15].

These multiple factors, combined with the rapid increase in the number of COVID-19 patients, led to a suspension of non-acute general healthcare in the Czech Republic. However, thanks to the effort of the Czech Dental Chamber and especially its members, dental care was provided throughout the whole pandemic, albeit with some restrictions. As the vast majority of the dental practices in the Czech Republic are private institutions, the anti-epidemic measures specific to dental practice were issued individually by each dental health care provider. Only a limited number of restrictions common for all public places were based on general government regulations, e.g., a limited number of people per area and disinfection being provided after entering interiors. Thus, the measures specific to dental practice were not issued centrally but were given at the individual discretion of the dentists. Such an individual approach was unusual in Europe and specific to the Czech Republic. However, to date, there are no statistics on the impact of the COVID-19 pandemic on dentistry in the Czech Republic. To reflect on it, the Czech Dental Chamber decided to perform a nationwide survey among its members, the results of which are presented in this article.

As membership in this professional chamber is mandatory for practicing dentistry in the Czech Republic, we were able to reach dentists from all around the country. A specific survey design based on a fast completion of the questionnaire resulted in an extraordinary response rate, representing, both relatively and absolutely, one of the most extensive sets of dentists as participants in a COVID-19 survey.

The aim of this work is to evaluate the impact of the COVID-19 pandemic on dental care in the Czech Republic from March 2020 to March 2021.

## 2. Materials and Methods

### 2.1. Design

This ad hoc, cross-sectional, self-administered, online survey was developed by the Czech Dental Chamber and performed among volunteers who were recruited from chamber members. All participants were informed about the objective of the study and did not have a patient status. The questionnaire was anonymous; completed questionnaires did not contain any personal information that could identify the participants and did not allow any traceability of the person answering. There were no direct benefits provided for participation. This study was conducted in accordance with the Declaration of Helsinki.

The survey consisted of 7 questions in the Czech native language, of which 6 were closed-ended, and 1 was open-ended. The questions were identified in cooperation with experts from the target population to ensure their clarity and appropriateness. The questionnaire was designed to be completed within 5 min.

The first question (Q1) regarded the impact of the pandemic on dental practices from March to May 2020. The dentists were asked about the opening hours of their practices, the number of patients treated, and the type of care provided. It was a closed-ended question with five single-choice answer options.

The second question (Q2) regarded the impact of the pandemic on dental practices from 1 September 2020, to 9 May 2021. The design was the same as in the first question. The first and the second questions followed the same design to reflect changes between the first and the second waves of COVID-19 in the Czech Republic. It was a closed-ended question with five single-choice answer options.

The third question (Q3) focused on the reasons for the closure of the dental practices during the whole pandemic. Respondents whose practices were not closed were asked to skip this question. It was a closed-ended question, five multiple-choice answer options.

In the fourth question (Q4), the participants were asked to evaluate the time range during which their dental practices were closed. Respondents whose practices were not closed were asked to skip this question. Six closed-ended answers option were provided. It was a closed-ended question with five single-choice answer options.

In the fifth question (Q5), the participants were asked to assess the patients’ interest in preventive care using the data retrieved from their dental practice management software. It was a closed-ended question with six single-choice answer options.

In the sixth question (Q6), the participants were asked to evaluate the impact of the pandemic on the oral health of their patients. It was a closed-ended question with four single-choice answer options.

The seventh question (Q7) regarded the participants’ greatest professional difficulty for the last year. It was an open-ended question with no limitations.

### 2.2. Sample

Officially registered e-mail addresses of 9922 Czech Dental Chamber members were addressed with a call for participation in this study focused on the impact of COVID-19 on practicing dentistry in the Czech Republic. Dentists were asked to participate in an online questionnaire from 24 February to 9 March 2021. As membership in this professional institution is mandatory for all practicing dentists in the country, the survey population included all practicing dentists in the Czech Republic. In this study, practicing dentists are defined as all dentists officially classified as Active members of the Czech Dental Chamber. According to the Czech Dental Chamber 2020 Annual Report, the chamber had 8624 Active members as of 31 December 2020 [16]. The rest of the 9922 e-mail addresses include members who are not practicing the dentist profession, including retired persons. Although these persons were not able to contribute to the survey, they are generally included in all full chamber correspondence.

### 2.3. Sample Size Relevancy

Considering the total number of practicing dentists in the Czech Republic, the minimum number of participants required for a relevant study was set to 368. The minimal sample size needed was calculated by Formula (1) using the online Netquest calculator. The calculation was performed with a study universe of all practicing dentists in the Czech Republic (N = 8624), a standard heterogeneity of 50%, a margin of error of 5%, and a confidence level of 95%. As the final sample size was 3674, the data within this study provide significantly relevant results.
(1)$$n=\frac{N\cdot Z^{2}\cdot p\cdot \left(1-p\right)}{\left(N-1\right)\cdot e^{2}+Z^{2}\cdot p\cdot \left(1-p\right)}$$

Formula (1). Relevant sample size calculation. Sample size calculated (*n*), size of the universe (*N*), deviation from the mean value (*Z*), the maximum margin of error tolerated (*e*), and the expected proportion (*p*).

### 2.4. Data Collection

The participants were contacted via e-mail invitation sent by the Czech Dental Chamber to officially registered the e-mail addresses of chamber members (*n* = 9922). The invitation included a link to an electronic questionnaire using a SurveyMonkey Questionnaire (SurveyMonkey, San Mateo, CA, USA). The questionnaire interface was compatible with a cell phone, desktop computer, or laptop to be accessible by as many respondents as possible. All replies were submitted fully completed. No exclusion criteria were applied. The participants’ responses were stored in the SurveyMonkey Questionnaire cloud database during the survey process and downloaded at its end.

### 2.5. Statistical Analysis

Results of closed-ended questions (Q1–6) were downloaded from the SurveyMonkey Questionnaire cloud database. The data representing numbers of responses were analyzed, and the results are presented as the percentage frequency of individual answers within all answers provided. Skipping of a question was not counted as an answer and was not included in the percentage pool.

Each response of Q7 with an open-ended setup was evaluated by the authors individually and classified into one of the 38 general categories. Answers including more than one piece of information were subclassified into several categories. Replies that did not occur repeatedly were included in the “Others” category. Answers that could not be evaluated because they had no content or the content was incomprehensible were included in the category “No reply/do not know/sorting not applicable”. Finally, the related categories were merged. The results of Q7 were processed as a table of the four most frequent categories summarizing the greatest professional difficulties among Czech dentists for the last year.

The data were analyzed using custom Microsoft Office Excel formulas (version 2106 for Windows, Microsoft Corporation, Redmond, WA, USA) and GraphPad Prism (version 8.0.0 for Windows, GraphPad Software, San Diego, CA, USA).

## 3. Results

### 3.1. Response Rate

Data from 3674 respondents were received. Compared to 9922 e-mails addresses included, the response rate was 37.0%. However, as this survey was addressed to practicing members (*n* = 8624), 3674 responses represent a 42.6% response rate within this population. Illustrated in Figure 1.

### 3.2. Impact of the Pandemic on Dental Practices from March to May 2020

Out of 3674 respondents, a significant majority (3334; 90.7%) stated that their practices were open in the spring of 2020. However, almost a quarter of them (788; 21.4%) were open only for urgent and acute cases. At the same time, nearly half of the respondents (1709; 46.5%) reported a decrease in the number of patients. Only a minority of respondents (141; 3.8%) stated that their practices were closed during the pandemic. Other impacts were stated in 201 (5.5%) replies (Figure 2).

These results show that almost all Czech dentists maintained at least basic care for patients, and nearly half of them reported a decrease in patients during the first wave of COVID-19 from March to May of 2020.

### 3.3. Impact of the Pandemic on Dental Practices from 1 September 2020, to 9 May 2021

Out of 3674 respondents, nearly all (3532; 96.1%) stated that their practices were open from 1 September 2020 to 9 May 2021. A few (31; 0.8%) restricted the care only for acute cases. Approximately one-third (1267; 34.5%) reported a decrease in number of patients. Almost no respondents (19; 0.5%) stated that their practices were closed (Figure 3).

These results show that practically all Czech dentists maintained care for patients during the first wave of COVID-19 from 1 September 2020 to 9 May 2021. A third of them reported a decrease in patient numbers.

### 3.4. Reasons Leading to Dental Practices’ Closure during the Whole Pandemic

This question was addressed only to the respondents whose practices were closed at any time during the pandemic. Respondents whose practices were not closed were asked to skip this question. Out of 3674 respondents, 1922 respondents replied to this question, and 1754 respondents skipped it. Multiple answers were allowed.

The 1922 respondents who responded to this question reported 2729 reasons leading to the closure of the dental practice. The results are presented as the number of answers, the percentage of respondents choosing this answer, and the frequency of an answer among all answers, respectively: Shortage of personal protective equipment: 970/50.5%/35.5%; COVID-19 outbreak in the workplace: 471/24.5%/17.3%; Fear of a possible self-infection: 461/24.0%/16.9%; Quarantine: 394/20.5%/14.4%; Other reasons: 433/22.5%/15.9% (Figure 4).

The data show that if the dental practices were closed, it was mainly due to the lack of personal protective equipment. Out of all reasons, shortage of protective equipment contributed to the closure of the dental practice in half of the cases, followed by the COVID-19 outbreak in the workplace and fear of a possible self-infection in a quarter of the cases, and quarantine in one-fifth of cases.

### 3.5. Time Range of Dental Practices Closure

This question was addressed only to the respondents whose practices were closed at any time during the pandemic. Respondents whose practices were not closed were asked to skip this question. Out of 3674 respondents, 1551 respondents replied to this question, and 2123 respondents skipped it.

Out of all participants, 1551 (42.2%) responded that their practices were closed during the pandemic. Half of these respondents (774; 49.9%) stated that the closure of their practices was in the range of 1–2 weeks. A total of 328 (21.2%) respondents reported the range of 2–4 weeks, and 152 (9.8%) reported the range longer than 1 month. The other reasons option was chosen by 297 (19.2%) respondents (Figure 5).

The results show that, during the pandemic, dental practices in the Czech Republic were mainly closed only for a short time, and closures longer than 1 month were rare.

### 3.6. Patients’ Interest in Preventive Care during the Pandemic

Out of 3674 respondents, 1498 (40,8%) stated that the patients’ interest in preventive care during the pandemic did not decrease at all or decreased by less than 10%, 1155 (31.4%) observed a 10–25% decrease, 474 (12.9%) observed a 26–50% decrease, 72 (2.0%) observed a 51–75% decrease, and 20 (0.5%) observed a >76% decrease, and 457 (12.4%) chose “No reply/do not know” option (Figure 6).

Most dentists observed no decrease or just a minor decrease in patients’ interest in preventive care during the pandemic.

### 3.7. Impact of the Pandemic on the Oral Care of Czech Patients

Out of 3674 respondents, 2852 (77.6%) did not observe any impact of the pandemic on the oral health of their patients, 621 respondents (16.9%) reported that their patients cared less about their oral health, and 61 respondents (1.7%) reported that their patients cared more about their oral health. The “others” option was chosen by 142 (3.9%) of respondents (Figure 7).

The approach to oral care was affected by the pandemic in less than one-fifth of the patients. For most of these, the effect was negative.

### 3.8. Dentists’ Greatest Professional Difficulties during the Pandemic

Out of 3674 responses, the most frequent professional difficulties were: difficulties with administration, reorganization, re-ordering due to illness and quarantine of patients, and, in general, crisis operating management (55%); concerns about the safety of staff and patients, difficulties in complying with the strict hygiene measures in the workplace (49%); shortage of personal protective equipment (21%); difficult to work in protective equipment (15%) (Figure 8).

## 4. Study Limitations

There are three major limitations in this study that could be addressed in future research. First, the presented study includes subjective reports of respondents. Second, the number of questions is limited. Third, some outcomes of this work are not comparable with the pre-pandemic period, as there are no studies or reports containing comparable data from the previous years. These limitations were not accidentally identified during or after the survey but were known to the authors before the research began.

This study provides an insight into the daily difficulties of dental professionals during the pandemic and helps identify the weaknesses of the healthcare system that are usually not retrievable via any other sources. On the other hand, some study outcomes, e.g., the complex quantification of the patients’ interest in preventive care (Q5) or impact of the pandemic on oral health (Q6), may be skewed as the quantification is based on the estimation of dental professionals. Although, naturally, the results of questionnaires are based on subjective judgments, we consider it important to comment on this issue and provide future direction for more accurate data acquisition. Q5 was included in the questionnaire mainly to retrieve qualitative results, i.e., whether the respondents observed a decrease in the interest in preventive dental care during the pandemic. The quantification may be considered abundant, the results may be questioned and should be verified by more objective data when available. More accurately quantified outcomes may be retrieved from government health institutions’ annual healthcare data reports once released.

Additionally, it is important to discuss the questionnaire extension itself. It may be argued that the questionnaire lacks some information that is commonly included in works of similar focus, such as participants’ personal data, dental practice location, or more questions, including those on COVID-19-related data. We agree that more questions would contribute to the informativeness of this study, but we intentionally decided to keep the questionnaire as simple as possible. This study was conducted at the time that the pandemic was escalating, and there were concerns that the respondents’ participation would be affected by their different workloads. It was assumed that those who worked during the pandemic had a reduced opportunity and determination to participate in a survey that consisted of many questions due to their extreme workload. Conversely, those who reduced or stopped their work during the pandemic were supposed to have a greater opportunity and determination to participate. Such an outcome would lead to an unequal representation of discrete groups of respondents and biased results. To address this issue, it was necessary to ensure that the survey is as inclusive as possible, i.e., with a time-saving and straightforward design that could be finished within 5 min. Alternatively, it was possible to wait with the survey for the time after the pandemic subsided. At this time, it could be assumed that dentists would have more time to respond. However, such an approach could notably affect the results, as respondents would be interviewed for events several months apart. In conclusion, the questionnaire was purposely developed to provide statistically robust and representative data at the cost of the data extension. This aim was achieved as 3674 participants represent 37.0% of all e-mails sent and 42.6% of practicing dentists in the Czech Republic.

As this study is the first of its kind in the Czech Republic, some data cannot be compared with data from the pre-pandemic period. For instance, the open/closed dental practices during the pandemic cannot be compared to the previous years. However, data of this study can be compared to the results of the studies carried out in the coming years.

## 5. Discussion

This study was carried out during the second wave of COVID-19 in the Czech Republic. The second wave was the most severe pandemic period regarding the number of new cases and deaths [17]. As there were no official data on the impact of the pandemic on Czech dentistry, the Czech Dental Chamber needed to obtain this information to adequately reflect on the needs of its members. Thus, the chamber decided to address dentists from all over the country with an online survey. As described in the previous chapter, the survey had to be specifically designed to address the broadest possible range of respondents. This unique approach focusing on a limited number of carefully selected questions led to a high participation rate, which accounted for 42.6% of practicing dentists in the Czech Republic. This outcome makes this study one of the largest in terms of the number of participating dentists and especially in terms of the percentage of participating dentists within one country. Furthermore, this is the first study on the COVID-19 impact on dentistry in the Czech Republic. However, although the minimum number of participants required for this study was significantly exceeded, it should be taken into account that the data presented in this study do not represent 100% of Czech dentists, but only those who participated in this study.

Q1 focused on the impact of COVID-19 on dental practices during the first wave of the pandemic (March–May 2020). The results revealed that although dental professionals were considered one of the most vulnerable frontline workers, more than 90% of dental practices were open [18]. Half of them registered fewer patients, about a fifth reduced the care provided only for acute cases, and about a fifth did not report any changes compared to before the COVID-19 outbreak. The number of closed dental practices was minimal. From these results, it is possible to conclude that during the first wave of the pandemic in the spring of 2020 in the Czech Republic:22.8% of dental practices were fully operational with no limitations compared to before the pandemic. The main operating limitation of dental practices (46.5%) was the lower number of patients.Acute dental care was provided almost without any restriction.Only 3.8% of dental practices were closed.

Q2 focused on the impact of COVID-19 on dental practices during the second pandemic wave (September 2020–May 2021). The percentage of open dental practices did not change significantly (from 90.7% during the first wave to 96.1% during the second wave). However, the results show an increase in dentists who worked without any changes compared to the period before the COVID-19 outbreak (from 22.8% during the first wave to 60.8% during the second wave). Fewer patients were reported by 34.5% of respondents, and only less than 1% of dentists treated only acute cases or closed their practice. It could be hypothesized that this change is probably due to the better availability of protective equipment, which has been identified as a major closure reason (see Q3), as well as a reduction in patients’ fear of infection. In summary, during the first wave of the pandemic in the spring of 2020 in the Czech Republic:Over 60% of dental practices were fully operational compared to before the pandemic; fewer patients were reported by 34.5%.Acute dental care was provided almost without any restriction.Only 0.5% of dental practices were closed.

Although these data suggest that access to dental care has not been notably limited, the real impact of a pandemic on oral health may be significant. As shown in the work of Nioi et al., some patients may have delayed treatment due to fear of infection, which could lead to worsening delays [19]. To assess these consequences, it is appropriate to choose a research tool other than a questionnaire, such as analyzing population oral health before, during, and after the pandemic.

In Q3, the dentists who closed their practices anytime during the pandemic were asked about the reasons for closure. The main reason was a shortage of personal protective equipment followed by a COVID-19 outbreak in the workplace, fear of infection, or quarantine. These reasons can be considered interlinked as the shortage of protective equipment may lead to infection, fear of infection, and quarantine. Indeed, most of the multiple answers included a shortage of protective equipment. This finding is in accordance with other studies reporting on the lack of protective equipment as one of the main issues of healthcare providers worldwide during the pandemic [20,21,22,23,24].

Of responders, 42.2% admitted that their practices were closed for some time during the pandemic. Out of these, 49.9% were closed for less than 2 weeks, 21.2% for less than 4 weeks, and 9.8% for more than 1 month (for any reason). These results show that, despite the pandemic, less than 10% of dental practices were closed for more than 1 month between March 2020 and March 2021. As the Czech Republic sets a minimum holiday of 4 weeks by law, Q4 results can be interpreted as showing that the pandemic did not notably extend the total time that dentists were out of work compared to the pre-pandemic years.

In Q5, respondents were asked about the decline in interest in preventive dental care they observed among Czech patients during the pandemic. The results show that 47.3% of dentists observed a decline greater than 10% compared to before the pandemic. As described in the Section 4, Q5 outcomes should be understood as a qualitative indicator of the interest in preventive dental care during the pandemic, and the quantification should be further verified. For accurate quantification, this issue can be addressed by studies covering healthcare data reports for years 2020 and 2021 once available and comparing them with previous years. Another interesting research direction could be the impact of lower interest in preventive dental care on the subsequent deterioration of oral health in the coming years. Additionally, in Q6, dentists were also asked whether they observed any impact of the pandemic on their patients’ oral care. More than three-quarters did not observe any impact, and poorer oral care of their patients was reported by 16.9% of dentists, and only 1.9% reported better oral care. Although almost 80% of dentists did not observe a deterioration in their patients’ oral care, almost a fifth did the opposite. Such an outcome can significantly contribute to the decline in overall oral health in the Czech Republic. Results of questions Q5 and Q6 indicate a worsening approach of Czech patients to oral health during the pandemic and may affect their oral health in the coming years. These outcomes may indicate the deterioration of oral health as an indirect outcome of the COVID-19 pandemic and serve as an incentive for government health institutions and the Czech Dental Chamber to introduce policies reversing this impact and promoting the oral health of the Czech population.

As revealed in Q7, the greatest professional difficulties of Czech dentists’ during May 2020–May 2021 were associated with the COVID-19 pandemic. Crisis operating management was the most significant inconvenience followed by the health concerns, lack of protective equipment, and problems resulting from working in the protective equipment. It is no surprise that all the abovementioned is linked to the pandemic as the dentists had to face this unprecedented situation from the frontline. These findings may indicate that the incidence of COVID-19 among dentists may be higher than in the general population. The notable impact of COVID-19 on the health of the healthcare workers was also reported by other authors; however, thus far, there are no such studies focused on Czech dentists [25]. This topic remains an opportunity for further investigation.

Additional studies with a similar focus reveal that the COVID-19 pandemic has severely affected dentistry globally. A work of Izzeti et al. reporting on dental activity during the pandemic in Italy shows that during the spring of 2020, 99.7% of surveyed dentists reduced their professional activity to urgent treatments or totally stopped working [26]. Another survey focusing on the COVID-19 impact on dental practices in Central Italy during the spring of 2020 reported on very similar trends [27]. In this study, 38.4% of respondents reported that they performed both dental emergencies and urgent care, 26.6% performed only urgent dental care, and 26.4% provided no dental care. In summary, 91.4% of respondents reduced their professional activities to urgent or emergent treatments or totally stopped working. This is in contrast to the reports of Czech dentists. In our survey, 22.8% of respondents admitted that their practices operated with no limitations, only 21.4% reported their practices remained open providing treatment only to acute cases, and 3.8% closed their practices during the spring of 2020. This disparity may be due to a different impact of the pandemic on Italy and the Czech Republic during these months. Italy was one of the most severely affected countries in the whole world; on the other hand, the Czech Republic was affected very mildly [28,29]. An additional study, performed by Miśta et al., focused on professional aspects of dentists in Poland during April 2020 [30]. A total of 71.2% of dentists responded that they decided to suspend their clinical practice. As in our study, the main reasons leading Polish dentists to practice closure were insufficient equipment (63.4%) and fear for their own and their family members’ health and life (51.2%, 57.6%, respectively). In June 2020, Ahmadi et al. performed a study among Iranian dentists assessing the pandemic impact on their practices [31]. Seventy percent of the participants admitted they did not perform non-emergency procedures during the pandemic, and 87% had problems finding and providing personal protective equipment. A study by Faccini et al. performed in May 2020 reports that in Brazil, 64.6% of the dentists performed only urgency/emergency dental care, 26.1% maintained routine appointments, and 9.3% closed their dental offices. These results are similar to those of our study. Although there are some methodological differences, our data show that, in comparison to surveys conducted in other countries, Czech dentistry remained more operative during the pandemic and Czech dentists faced similar fears and problems as their colleagues from other countries.

## 6. Conclusions

This survey conducted among 3674 dentists in the Czech Republic reveals that more than 90% of them worked during the COVID-19 pandemic from March 2020 to March 2021. Out of those who closed their practices, only 9.8% was for more than 1 month. The main reasons leading to the closure were a shortage of protective equipment, COVID-19 infection, fear of infection, or quarantine. The leading professional difficulties of Czech dentists were crisis operating management, a lack of protective equipment, and health safety concerns. Additionally, the respondents observed a declining interest in preventive dental care in 47.3% of patients and worse oral care in 16.9% of patients indicating a deterioration in oral health as a possible delayed outcome of the pandemic.

## Figures and Tables

**Figure 1 ijerph-18-09121-f001:**
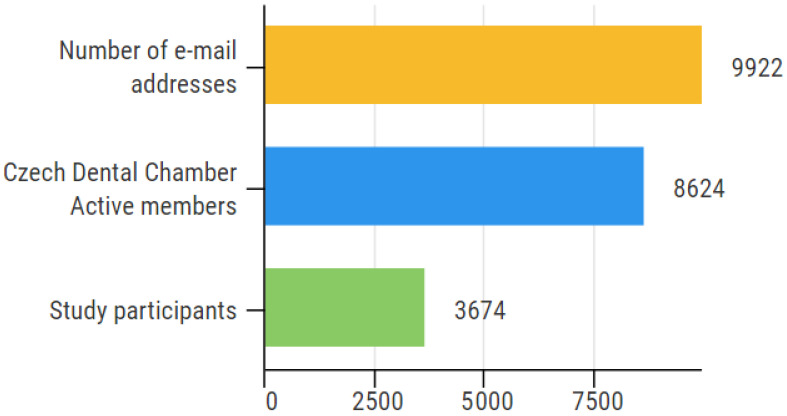
Response rate: 3674 participants represent 37.0% of all e-mail addresses included and 42.6% of practicing dentists in the Czech Republic.

**Figure 2 ijerph-18-09121-f002:**
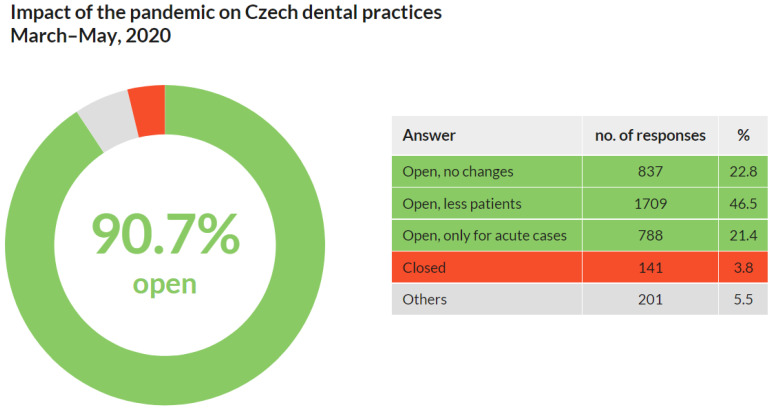
Impact of the pandemic on Czech dental practices from March to May of 2020. More than 90% of them remained open.

**Figure 3 ijerph-18-09121-f003:**
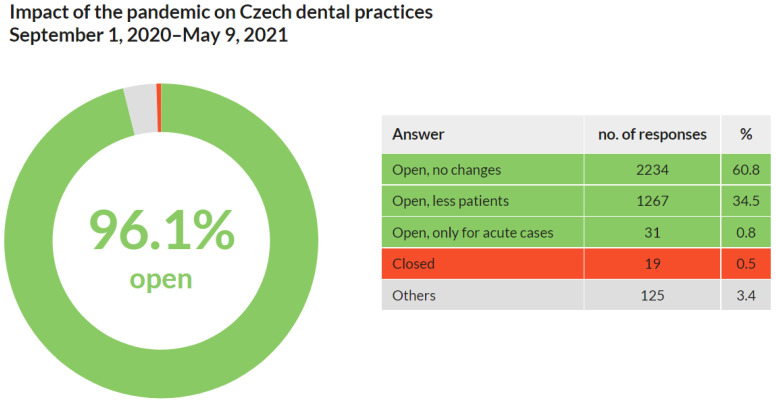
Impact of the pandemic on Czech dental practices from 1 September 2020 to 9 May 2021. More than 96% of them remained open.

**Figure 4 ijerph-18-09121-f004:**
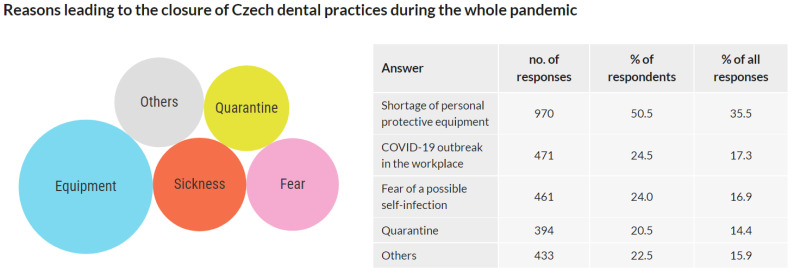
Reasons leading to the closure of Czech dental practices during the whole pandemic.

**Figure 5 ijerph-18-09121-f005:**
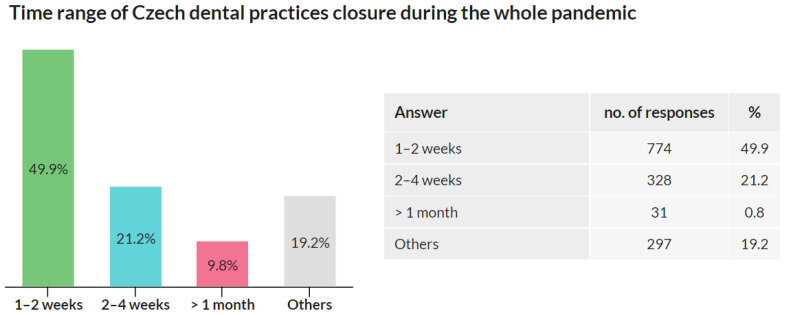
Time range of Czech dental practices closure during the whole pandemic.

**Figure 6 ijerph-18-09121-f006:**
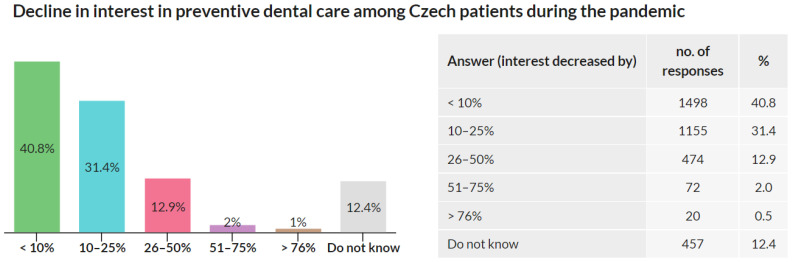
Decline in interest in preventive dental care among Czech patients during the pandemic.

**Figure 7 ijerph-18-09121-f007:**
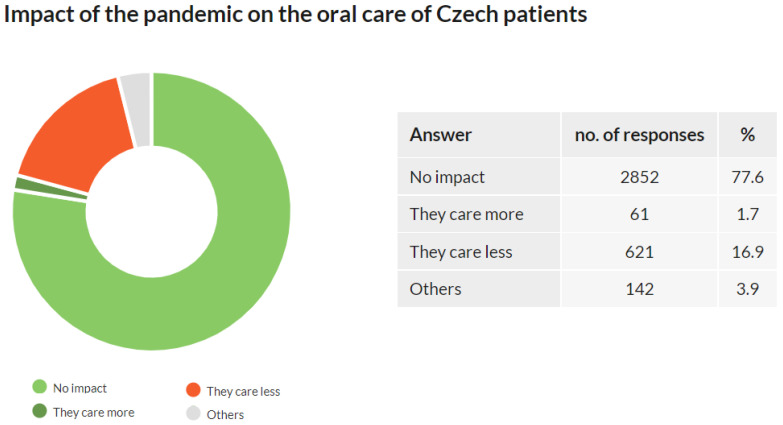
Impact of the pandemic on the oral care of Czech patients.

**Figure 8 ijerph-18-09121-f008:**
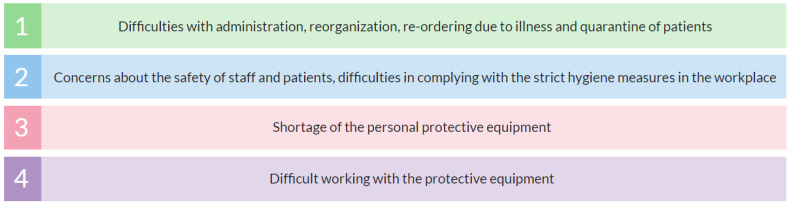
Czech dentists’ greatest professional difficulties during the pandemic.

**Table 1 ijerph-18-09121-t001:** Overview of selected key anti-pandemic measures issued by the Czech government.

	Issued on	Effective from	Description of Key Measures
2020	March 12	March 12	Declaration of the state of emergency in the Czech Republic
	March 13	March 14	Prohibition of selected leisure activities; prohibition of retail sales and services
	March 15	March 16	Prohibition of free movement of persons (with exceptions, including healthcare consumption)
	March 18	March 19	Mandatory covering of mouth and nose indoors and outdoors
	April 30	May 18	End of the emergency
	September 30	October 5	Declaration of the state of emergency in the Czech Republic
	September 30	October 5	Prohibition of selected leisure activities
	October 12	October 13	Obligation to use face masks indoors
	October 19	October 21	Obligation to use face masks indoors and outdoors
	October 21	October 22	Prohibition of free movement of persons (with exceptions, including healthcare consumption)
	October 21	October 22	Prohibition of retail sales and services
2021	February 26	April 11	End of the emergency

## Data Availability

The dataset is available on-demand from the corresponding author.

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
