# Peer review of "Impact of COVID-19 on Czech Dentistry: A Nationwide Cross-Sectional Preliminary Study among Dentists in the Czech Republic"

_ijerph, 2021, doi:10.3390/ijerph18179121_

Round 1

Reviewer 1 Report

The study is really interesting and touching contemporary topic. 

However introduction lacks of references (I would have mentioned the protcols adopted by dentists allowed the low rate of contgiosity for example) and data are not discussed with similar studies carried in other countries (https://bmcoralhealth.biomedcentral.com/articles/10.1186/s12903-020-01273-6; https://onlinelibrary.wiley.com/doi/10.1111/odi.13606; https://onlinelibrary.wiley.com/doi/full/10.1111/clr.13676; https://pubmed.ncbi.nlm.nih.gov/33272261/; https://www.mdpi.com/1660-4601/17/16/5780; https://www.mdpi.com/2227-9032/9/4/454) 

Reviewer 2 Report

Dear Colleagues!
Thank you for an interesting study that sufficiently reveals the problem of adapting to new working conditions during a pandemic. I read it with great interest, both as a medical practitioner and as a scientist.
However, despite this interest, I had a question about the title of the work: is it really necessary to discriminate against those who were not included in the study, indicating only the percentage of respondents? It seems to me more correct to remove the percentage of participants from the title of the article and talk about it only in the section Materials and Methods.

At the same time, on line 128, you will sew about the number of participants in the study, but the percentage of the name remains unclear. It is necessary to clarify the number of participants in the study and those who did not participate in it. So it may seem that among those about whom you write in line 128-130 make up 52.5%, which significantly exceeds the number of working specialists, creating an obvious shortage of personnel.

In general, the work was carried out at a good methodological level, however, it is necessary to more detailed the methods of statistical data processing, as well as the criteria for participation: was it anonymous or the respondents were obliged to disclose their data. Discussing the results obtained, it seems to me useful to take into account the fact that less than half of the country's doctors were included in the study, and one should rely on the percentage data in comparison only with an indication of the lack of information about the majority, which was not included in the study.

Reviewer 3 Report

I have read with great interest the work entitled “ Impact of COVID-19 on Czech dentistry: a nationwide cross- sectional study among 47.5% of practicing dentists in the Czech Republic”

The work proposes investigate  the impact of the COVID-19 pandemic on dental care in the Czech Republic from March 2020 to March 2021

Major Concerns

  1. Not all governments have taken the same measures for the pandemic. I ask the authors in the introduction for a brief summary of the measures taken in the Czech Republic summarizing it  in a figure. 
  2. Were there many infections among dentists in the Czech Republic? In other countries they are among the most affected categories. Authors can take as an example and compare the data with Nioi, Matteo, et al. "COVID-19 and Italian healthcare workers from the initial sacrifice to the mRNA vaccine: Pandemic chrono-history, epidemiological data, Ethical Dilemmas, and Future Challenges." Frontiers in Public Health 8 (2020).
  3. Is dentistry in the Czech Republic entrusted exclusively to individuals or is there a part of the activity guaranteed by public structures?
  4. In some states, a criminal shield for medical and dental liability has not been proposed for those who operated during COVID-19 (in the case of dentists, culpable transmission of the disease was at stake). I would like the authors to briefly talk about the problem in the introduction by comparing it with "COVID-19 and medical liability: Italy denies the shield to its heroes." EClinicalMedicine 25 (2020).
  5. The data concerning openings / closings is interesting but should be compared with a period in which there was no pandemic (i.e., 2019). If the authors do not have this possibility, this problem must be indicated among the limits of the study.
  6. As regards the reduction of activities, the result of the questionnaire should be compared to the real numbers. The one reported is in fact only the impression of the professionals who responded. Do the authors have any concrete data on this? They believe that what 60% said is plausible, namely that there has not been a decline in access (compare with "Fear of the COVID-19 and medical liability. Insights from a series of 130 consecutives medico-legal claims evaluated in a single institution during SARS-CoV-2-related pandemic." (2021).
  7. Is the questionnaire validated? In the case of a preliminary study, the title should be adequate.
  8. Has an estimate of the sample size been made?
  9. References number isn’t sufficient. Please expand it.

Overall, I liked the article and I am confident that the authors can quickly meet the few corrections required in order to speed up the editorial evaluation.

Round 2

Reviewer 3 Report

The authors have made the required extensions and I think the paper in the current version has greatly improved. I have no further observations to make.